

# Using seasonal forecasts to enhance our understanding of extreme European windstorm impacts

Jacob W. Maddison[1], Jennifer L. Catto[1], Sandra Hansen[2], Ching Ho Justin Ng[2], and Stefan Siegert[1]

[1]Department of Mathematics and Statistics, University of Exeter, Exeter, UK
[2]Guy Carpenter & Company Limited, London, UK

**Correspondence:** J.W. Maddison (j.maddison2@exeter.ac.uk)

**Abstract.** Considerable effort is spent at insurance and reinsurance companies to estimate the risk posed by windstorms. Among these risks, strong near surface wind speeds can be particularly damaging, threatening infrastructure, human life, and billions of pounds in insured losses. Here, we use nearly 700 years worth of extended wintertime seasonal forecast output to estimate the impact of extreme European windstorms, with insured losses estimated using a storm severity index (SSI).

Using the full integration period of the seasonal forecast model, we follow the UNprecedented Simulated Extreme ENsemble (UNSEEN) method, here applied to windstorms for the first time. After demonstrating that the seasonal forecast model of the UK Met Office represents windstorms with good accuracy, and developing a new method to convert from wind speed to wind gust derived SSIs, the likelihood of occurrence of unprecedented windstorms is quantified for several countries within Europe. The probability that a windstorm that impacts a country will be more extreme than any observed (i.e. an unprecedented or

unseen windstorm) is generally between 0.5% and 1.6%. The North Atlantic Oscillation (NAO) is shown to influence European windstorms: strongly positive and negative NAO values strongly increase and decrease the likelihood of an unprecedented storm, respectively.. Serial clustering of windstorms within an extended winter is also found to increase the aggregated seasonal impact of windstorms for the countries analysed herein. These results may aid in the prediction of seasonal loss totals, as the NAO, for example, is predictable several months in advance. The analyses presented could be extended to other datasets, thus

increasing the sample size of windstorms and allowing for the estimation of very high return period storms and the potential losses insurance companies will be liable to cover.

## 1 Introduction

Widespread occurrences of strong winds, also known as windstorms and commonly associated with extratropical cyclones, are among the most damaging natural hazards in Europe and therefore one of the most costly (Schwierz et al., 2010; Munich Re,

2015). Costs are incurred, into the billions of pounds, when insurance claims are made against damaged (insured) property. The potential impacts of windstorms are therefore of great interest to insurance and reinsurance companies. In light of this, much research effort has been devoted to quantifying how much monetary loss historical storms have caused, and what may be expected from storms in the future (e.g. Klawa and Ulbrich, 2003; Leckebusch et al., 2007; Schwierz et al., 2010; Donat et al., 2011; PERILS AG, 2022). Of great interest to the insurance sector are extreme windstorm events, or those that occur



at high return periods (and in particular the one in two hundred year events that must be covered in line with the Solvency II directive [1]). To study the statistics of these extreme storms, and estimate their impacts, many hundreds, or even thousands, of years worth of data are needed. In addition, the data must contain realistic representations of windstorms. In this paper, the losses that may be expected from damage due to strong winds in the most extreme extratropical cyclones (up to a one in around seven hundred year event) are quantified for several countries within Europe using a large dataset.

The need for a dataset that extends over several hundreds of years, and that is of the spatial and temporal resolutions required to sufficiently represent a particular feature of interest, is ubiquitous in climate science. Observations may be available that cover the extended time period for certain variables (e.g. temperature), but lack the spatial resolution and coverage normally required, whilst reanalysis products offer (relatively) high spatial and temporal resolution and include many variables, but are limited in the length of their record (typically less than 100 years). Model data is thus often the only option for obtaining

such a dataset. In relation to windstorms, climate model output may be too low resolution for reliable cyclone-related surface wind speed information and, due to the long integration of climate model simulations, may contain considerable biases (e.g. Donat et al., 2011; Tian et al., 2019; Priestley and Catto, 2022; Miao et al., 2023). In catastrophe models, climate model data is downscaled, either dynamically or statistically, to account for this and obtain the necessary resolution to study windstorm impacts. The UNprecedented Simulated Extremes using ENsembles (UNSEEN) approach (Thompson et al., 2017) is an alter-

native for creating the desired dataset: output from many simulations from seasonal forecast models (8 of which are available through the Copernicus climate data store (CDS, 2023)) are combined to produce a large dataset of model output. As the model is run in seasonal forecast mode, i.e. initialised from reanalysis data each season, run at higher-resolution (compared to typical climate model integrations) and for shorter integration periods, the model output is taken to represent an observation-like data set. The key assumption of the approach is that seasonal forecast systems simulate the feature of interest accurately enough

to be considered as an alternative to reality. Previous studies have taken this approach to successfully quantify extremes in, for example, rainfall (Thompson et al., 2017; Kelder et al., 2020; Kent et al., 2022), temperature (Thompson et al., 2019; Kay et al., 2020) and drought (Squire et al., 2021).

    Windstorms are another potentially suitable candidate for applying the UNSEEN methodology: there exists some skill in predictions of the frequency and intensity of windstorms in seasonal forecast models (Scaife et al., 2014; Befort et al., 2019;

Degenhardt et al., 2023). Indeed, an event set of European windstorms was derived from seasonal forecast data from the ensemble prediction system (EPS) of the European Centre for Medium Range Weather forecasts (ECMWF) in Osinski et al. (2016), an example of the UNSEEN methodology before it was so named. The seasonal forecast system was shown to produce storms with characteristics similar to a reanalysis dataset and hence suitable for creating a large sample of realistic windstorms and facilitating the study of very rare events. The dataset produced in Osinski et al. (2016) was made using 15-day integrations

of the ECMWF-EPS (Molteni et al., 1996; Palmer et al., 2007), which at the time was not coupled to an ocean model. As such, the windstorms produced are not independent from observations and are somewhat constrained to the climatology of the period (e.g. the SSTs). Of course, modifications to windstorm properties and their statistics are found due to the 300-fold

---

[1]The European Union Solvency II directive dictates losses must be covered with 99.5% confidence (https://eur-lex.europa.eu/legal-content/EN/ALL/?uri=CELEX:32009L0138 [last access February 2024]



increase in sample size. More recently, Lockwood et al. (2022) used output from a higher resolution global climate model to create a 1300-year event set of winter windstorms for Europe, again finding good agreement with reanalysis in the number of storms per season and the total storm losses within a season. Several previous studies have also used climate models to estimate extreme storm impacts for individual countries (Klawa and Ulbrich, 2003; Karremann et al., 2014), and the European region as a whole (Priestley et al., 2018). Here, we add to this growing body of research by tracking extratropical cyclones in 672 extended winter seasons worth of Met Office seasonal forecast data, and estimating their impacts using a loss index. This study goes beyond that previously reported in the literature and presents new results by:

1. for the first time using 215-day integrations of a fully coupled seasonal forecast model (in contrast to the 15-day integrations of mostly uncoupled seasonal forecast data used in Osinski et al. (2016)). This allows for the creation of a windstorm dataset that is more independent from observations and which may include very extreme storms that potentially could be observed in the real world today (providing the model does not drift too far from reality during its integration).

2. taking the UNSEEN approach in the context of European windstorms, that is focusing on *unprecedented* windstorms and quantifying their impacts and likelihood for several European countries. This is in contrast to previous work (e.g. Karremann et al., 2014; Osinski et al., 2016; Priestley et al., 2018; Lockwood et al., 2022) where the results were presented in terms of storm return period and not related to the most extreme event in the historical record.

3. producing results that are directly applicable for end users, e.g. (re)insurance companies. Here, we estimate storm impacts that would be expected from extreme wind gusts, and perform the majority of analyses at a country-by-country level. This allows for direct comparison with the catastrophe models developed and used by (re)insurance companies (that are based on wind gust and typically aggregated to country level) and can be used in their evaluation and development.

Estimating windstorm impacts from model data has been the focus of much research. Due to the confidential nature of windstorm losses (i.e. how much money an insurance or reinsurance company paid out as a result of the damage from a particular windstorm), publicly-available monetary loss data is scarce. In lieu of this, loss indices (often termed storm severity indices (SSIs)) have been introduced and used to estimate losses based on the winds, typically the maximum wind gust, associated with a given windstorm (e.g. Klawa and Ulbrich, 2003; Leckebusch et al., 2008). SSIs have been shown to correlate well with actual loss data for certain regions within Europe (Klawa and Ulbrich, 2003), or Europe as a whole (Little et al., 2023), and are used routinely by insurers and reinsurers as a proxy for insured losses. The aggregated losses caused by cyclones over a given period, for example the winter season (when cyclones are typically stronger and damages are greater (Munich Re, 2017; Kron et al., 2019)), is of particular interest to the insurance sector. Features of the large-scale atmospheric circulation can impact the aggregated losses in a season. Increased clustering of extratropical cyclones, i.e. occasions when multiple cyclones pass over a region in quick succession, results in higher seasonal loss values from medium-to-high return period storms (Priestley et al., 2018). Fluctuations in the North Atlantic Oscillation (NAO) or East Atlantic pattern (EA), also impact seasonal losses due to windstorms (Lockwood et al., 2022), as well as the seasonal prediction of the windstorms themselves (Degenhardt



et al., 2023). Further quantification of the relationship between the large-scale atmospheric circulation and seasonal loss totals is beneficial for the insurance industry, as it may offer a route to extended predictability of seasonal losses, and is included herein.

There are three main aims of the work presented in this paper.

– To produce a dataset of extratropical cyclone tracks and their associated wind footprints and SSI values that contains several hundreds years worth of data and can therefore be used to study extreme (or high return period) storms.

    – To use this dataset to gain insight into the potential impacts of higher return period storms and how they compare to well-known damaging storms from the recent past, and estimate the probability of unprecedented storms.

    – To quantify the impact of the large-scale atmospheric circulation, in particular cyclone clustering and the NAO, on the

100       occurrence of extreme windstorms and their induced losses.

The article is organised as follows. In section 2, the data underlying the analyses presented are described. The methods that are followed herein, including the cyclone tracking, SSI calculation and bias correction, are detailed in section 3. The model fidelity to follow the UNSEEN approach is assessed in section 4. In section 5, the characteristics of windstorms are compared between the seasonal forecast data and a reanalysis product, as well as quantifying how they differ for the most

extreme storms. Return periods of windstorm impacts are calculated in section 6, as well as how cyclone clustering impacts storm return periods. In section 7, the likelihood that an unprecedented windstorm impacts a country is calculated, as well as an estimation of the influence of the NAO on unprecedented windstorms. The article is concluded in section 8.

## 2   Data

The primary data used in this study are from a seasonal forecast system that was rerun for 24 years in the past (i.e. seasonal

hindcasts). The seasonal hindcasts used are from the Global Seasonal Forecast System version 6 (GloSea6-GC3.2 system 601, Williams et al., 2018), rerun at the Met Office for years 1993–2016. Fields on a regular Gaussian grid (F128, i.e. a full Gaussian grid with 128 latitude lines between the pole and equator, which is approximately a $0.7° \times 0.7°$ latitude-longitude grid) are retrieved for analysis from the Copernicus Climate Data Store (CDS, 2023). Instantaneous values of mean sea level pressure and horizontal wind components at 10 m are downloaded every 6 hours and used in the tracking algorithm and SSI

calculation, respectively (described in sections 3.1 and 3.2). The seasonal hindcasts initialised in September (on 01, 09, 17 and 25 September of each year), and run for 215 days, are used herein for each perturbed ensemble member (7 members total). The hindcast runs therefore cover the extended winter season (October–March). In total, 672 extended winter seasons are analysed (4 hindcast initiation dates, 24 hindcast years, and 7 ensemble members) with a total of approximately 120,000 storms identified in the North-Atlantic/European regions (see section 3).

For verification and bias correction purposes, the fifth generation atmospheric reanalysis from the European Centre for Medium-range Weather Forecasts (ERA5, Hersbach et al., 2020) is used. Meridional and zonal wind components at 10 m, as well as the maximum 10-m wind gust since previous post processing, are downloaded for calculating the SSI. Mean sea level



pressure is downloaded for verification of the storm tracks. Data are retrieved for the years covered by the GloSea6 hindcasts (1993–2016) and at the same spatial and temporal resolutions, namely F128 and 6 hourly.

## 3 Methods

### 3.1 Cyclone tracking

Extratropical cyclone tracks are produced using the objective feature tracking algorithm TRACK of Hodges (1994, 1995, 1999). Mean sea level pressure is used to identify and track cyclones, as is done in Hoskins and Hodges (2002), with minima in the pressure field (i.e. low pressure systems) identified and tracked at a temporal resolution of 6 hours. Prior to tracking, noise

is reduced in the field by truncating it to T63 (triangular truncation with a maximum wavenumber of 63) and planetary-scale features are removed (those with total wave numbers less than or equal to 5). Extrema in the preprocessed mean sea level field are identified and then joined together to form system tracks. Here, the technique of Hodges (1995) is used to create the tracks, in which a cost function of the ensemble track smoothness is minimised to obtain the ideal set of tracks. This approach includes constraints on the smoothness of the tracks and the maximum displacement between adjacent track points (Hodges, 1999). For

this work, only tracks that last for longer than two days and travel further than 1000 km are retained, as these are the more mobile systems expected to generate greater impact.

Wind field footprints are created for each storm track in the seasonal forecast data as follows. First, the wind field is masked in a circle of radius 25° centred on each point of a cyclone's track. Then, the maximum wind at each grid point within the storm footprint across the storm life cycle is retained. This constitutes the storm footprint that is used in the calculation of the

SSI.

### 3.2 Storm severity index and bias correction

The storm severity index (SSI) based on that introduced in Klawa and Ulbrich (2003) is used here, in accordance with the vendor catastrophe models typically used in the insurance and reinsurance sector. The SSI is calculated using the formula

$$SSI = \sqrt[3]{\frac{\sum_{i \in \Omega_0} (v_i - v_0)^3 A_i}{\sum_{j \in \Omega} A_j}}, \tag{1}$$

where $v_i$ and $v_0$ are the wind field at the $i^{th}$ grid point and the wind threshold that defines extreme winds, respectively. The $\Omega$ denotes the set of all grid points corresponding to the storm footprint, and $\Omega_0$ denotes the subset of grid points where $v_i > v_0$, and $A_{i/j}$ the area of the $i/j^{th}$ grid point. The cube of the wind speeds is used in the impact metric as the cube of the wind speed is proportional to the flux of kinetic energy, and is therefore expected to introduce a strong wind speed-loss relationship (Klawa and Ulbrich, 2003). Thus, the SSI represents both the intensity of the damaging gusts and the geographical extent of

the storm.





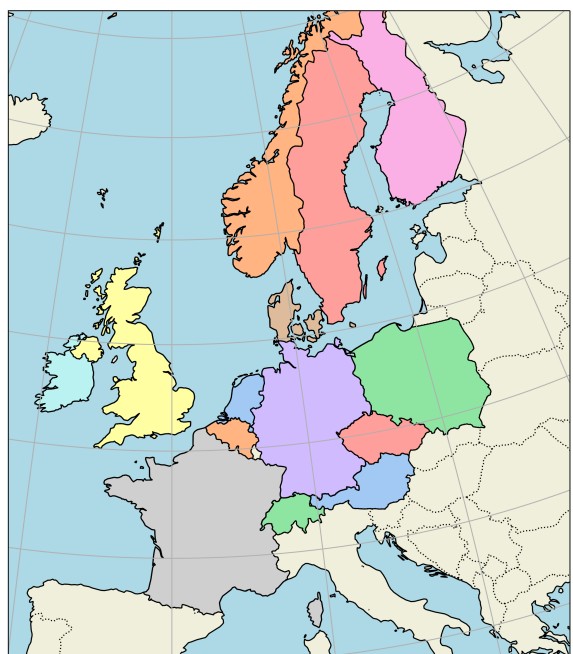

**Figure 1.** The European countries that are included in the analysis presented in this article.

The SSI can be used to aggregate losses over any specified region and time period. Here, the SSI is calculated at the country level for each storm track footprint, as well as aggregated over several countries to provide a European-level SSI value. The countries (and their abbreviations) included here (shown in Figure 1) are Austria (AT), Belgium (BE), Switzerland (CH), Czechia (CZ), Germany (DE), Denmark (DK), France (FR), Finland (FI), Great Britain (GB), Ireland (IE), Norway (NO), Netherlands (NL), Poland (PL) and Sweden (SE), with the European-level SSI an aggregation across these countries. Note that Luxembourg does not appear in the analyses as it is smaller than the F128 grid scale used here. The countries included cover northwestern Europe (Fig. 1), the region closest to the jet exit (the region where most extratropical cyclones impact Europe). An SSI value using the cyclone wind footprint as input and following equation 1 is calculated for each individual storm track. The storm footprint SSIs are calculated using the wind gust and wind speed fields in ERA5, and the wind speed fields in GloSea6. The SSIs for each storm footprint in a season are summed to measure the total loss for an extended winter season (termed the annual exceedance probability (AEP) here, following industry conventions). Finally, the wind speed based SSIs are bias corrected to values representative of wind gust based SSIs (see next section).

### 3.2.1 Conversion from wind speed SSI to wind gust SSI

SSIs are typically calculated using the maximum surface wind *gust* (e.g. Klawa and Ulbrich, 2003). As these are not available from the GloSea6 archive, a method is developed to calculate SSIs from surface wind *speed* data and convert them to values representative of wind gust SSIs. Recently, Lockwood et al. (2022) similarly estimated wind gust SSIs using the wind speed



output from a climate model. Their approach involved bias correcting the wind speed field to a wind gust field and then calculating an SSI. The approach used here first calculates an SSI using the wind speed field and then bias corrects to a wind gust SSI. This has the advantage of requiring less data processing to produce the bias correction, which is useful for such large

datasets.

The SSI is intended to identify damage inducing winds on approximately 2% of days, following Klawa and Ulbrich (2003). The 20 m/s wind gust threshold value used in the wind-gust SSI in this and other studies was defined based on the 2% of days criteria for wind gust values and insurance company practises in Germany (Klawa and Ulbrich, 2003), but has been widely adopted as a suitable value to use for other European countries and is the threshold commonly used in catastrophe

models run by insurance companies. To calculate an SSI from wind speeds it is therefore required to define a wind speed threshold for the SSI that is equivalent to the 20 m/s wind gust threshold. To do this, the percentile that 20 m/s represents in the ERA5 wind gust climatology is identified, and then the wind speed threshold ($v_0$) is defined as the wind speed value at that percentile in the climatological wind speed distribution. A 20 m/s wind gust is typically between the $96^{th}$ and $98^{th}$ percentile for the countries considered, with a mean of the $97.5^{th}$, (supporting the use of a 20 m/s wind gust threshold across

countries as done in catastrophe models). The wind speed thresholds used in the wind speed SSI are calculated separately for each country considered to account for potentially different wind gust-wind speed relationships across countries, as well as in ERA5 and GloSea6 separately to account for potential GloSea6 biases in near surface wind speed. The threshold values used for calculating the wind speed SSI in ERA5 and GloSea6 are listed in Table 1, together with the standard error in the threshold value estimate.

| Country | AT | BE | CH | DE | DK | FI | FR |
|---|---|---|---|---|---|---|---|
| ERA5 | 6.1 ±0.4 | 9.0 ±0.5 | 8.5 ±0.5 | 8.4 ±0.4 | 9.5 ±0.6 | 10.0 ±1.1 | 8.5 ±0.4 |
| GloSea6 | 5.9 ±0.4 | 9.8 ±0.5 | 8.8 ±0.5 | 9.7 ±0.5 | 11.7 ±0.6 | 9.8 ±0.4 | 9.5 ±0.4 |
| Country | GB | IE | NL | NO | PL | SE | |
| ERA5 | 8.9 ±0.5 | 8.4 ±0.5 | 9.4 ±0.7 | 6.4 ±0.5 | 9.4 ±0.7 | 8.2 ±0.5 | |
| GloSea6 | 10.3 ±0.4 | 10.8 ±0.4 | 11.0 ±0.5 | 8.4 ±0.6 | 10.1 ±0.6 | 8.9 ±0.4 | |

**Table 1.** Threshold values ($v_0$, m/s) used in the calculation of the wind speed SSI together with the standard error of the estimate of the percentile used in the threshold calculation. Values are shown for ERA5 and GloSea6. Standard errors are calculated as the standard deviation of wind speed threshold values calculated across evenly sized samples (1000 data points) of the data.

The final step in the procedure for converting SSI values derived from wind speeds to values representative of wind gust SSIs is bias correction. This correction is required for the data produced to be used in the evaluation and development of catastrophe models, and for a better comparison with the previous literature. A quantile-mapping bias correction approach (Thrasher et al., 2012) is used here to map the wind speed SSIs to wind gust SSIs. Quantile maps are made for each country in Europe using the ERA5 wind gust SSI and ERA5 wind speed SSI. The approach works as follows. First, cumulative distribution functions

(CDFs) of the model (ERA5 wind speed SSI) and reference (ERA5 wind gust SSI) data are estimated over a set of regularly spaced quantile intervals. Then, to bias correct a wind speed SSI value $x$, the percentile of $x$ is calculated in the model CDF





estimate. The value in the reference CDF at this percentile is found and taken as the bias corrected wind speed SSI, i.e. the wind gust SSI estimate. The ERA5 quantile maps are used for both the ERA5 wind speed SSIs and GloSea6 wind speed SSIs to enable the study of extreme SSI values using GloSea6. A quantile map by definition maps the extremes of the model data to

the reference data and thus the bias corrected values cannot exceed the maximum value in the reference data on which they are made (i.e. the maximum ERA5 wind gust SSI). By using the ERA5 based quantile maps for GloSea6, we retain the possibility that bias corrected SSI values in GloSea6 can exceed the maximum in ERA5 (when the GloSea6 wind speed exceeds the maximum ERA5 wind speed).

### 3.3 North Atlantic Oscillation index

The North Atlantic oscillation (NAO) index used here is based on the average mean sea level pressure difference between Iceland (defined here as within the region $63°–70°$N, $16°–25°$W) and the Azores ($36°–40°$N, $20°–28°$W). The average pressure over the Azores is subtracted from that over Iceland and therefore positive values represent the positive phase of the NAO (NAO+) and a strong pressure gradient over the North Atlantic. NAO index values are calculated at each (6 hour) time step in the GloSea6 hindcasts and the corresponding times in ERA5.

## 4  Model fidelity

The GloSea6 seasonal forecast system is first tested in its ability to reproduce key characteristics of windstorms to verify that its output is viable for the purpose of this study.

### 4.1  The North Atlantic storm track

Two key features of the North Atlantic storm track are compared in the GloSea6 hindcasts and ERA5 reanalysis in Figure 2:
the average number of storms passing through each grid point across the extended winter (track density, Fig. 2 left panels), and the average pressure anomalies of those storms (mean intensity, Fig. 2 middle panels). In GloSea6, the well known features of the North Atlantic storm track seen in ERA5 are reproduced. Storms are most frequent over the Gulf Stream region (extending from Newfoundland towards Iceland and northwest Europe), with a gradual reduction of storm frequency to the south and a sharp reduction to the north associated with Greenland. The number of storms per season is above 60 for much of this region
in both GloSea6 and ERA5. Accordingly, the bias in track density is generally small (Fig. 2(g)) for the majority of the North Atlantic and European continent. The largest biases are found in the jet exit region, with the pattern in the bias suggesting the storm track is less tilted in GloSea6 than ERA5 (a bias common in climate model simulations (Pithan et al., 2016)). The bias in track density in GloSea6 is small for the majority of the countries considered here (Fig. 1(g)). Considering the bias in track density relative to the value in ERA5 reveals biases reveals a region of positive relative biases of around 25% in the region of
positive bias extending from Scotland to Denmark (Fig. 2(g)) and negative relative bias of -20% over Scandinavia. The mean intensity of storms is also in good agreement in GloSea6 and ERA5, with the intensity of storms highest in the centre of the storm track region (Figs. 2(b),(e)). There is a marginal overestimation of intensity in northwestern Europe (Fig. 2(h)), but the



bias in the mean intensity of storms is generally small. Relative to the value in ERA5, the relative bias in mean intensity is less than 5% for most of the region considered, expect in northwest France and the southeast of Great Britain where the relative

bias is positive and approximately 15%.

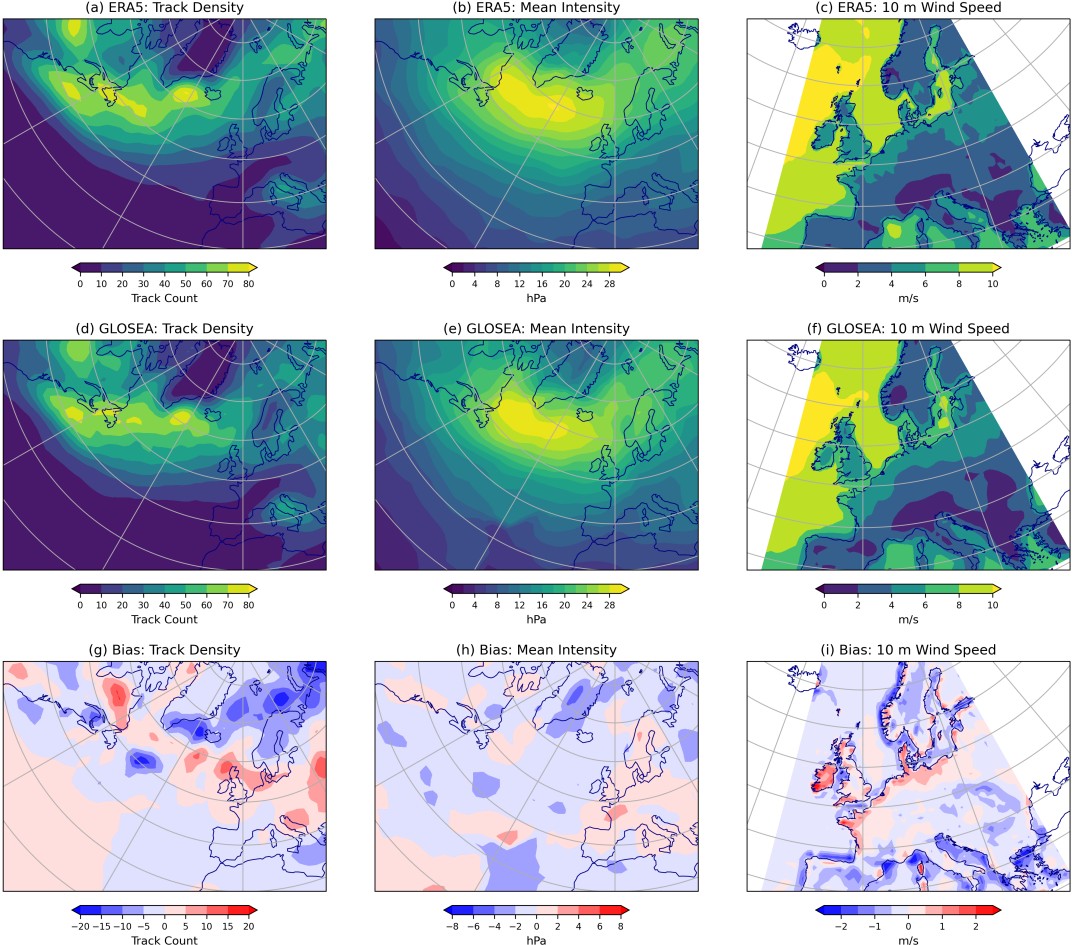

**Figure 2.** A comparison of the GloSea6 seasonal forecast dataset and the ERA5 reanalysis for: the number of storms identified in each grid point during the extended winter (left hand panels), the mean intensity of these storms (where intensity is measured as the magnitude of the central pressure anomaly, middle panels), and the 10 m wind speed (right hand panels). Fields are shown for ERA5 (top row), GloSea6 (middle row), and their difference (GloSea6 minus ERA5, bottom row).

The 10 m wind speeds are also compared in GloSea6 and ERA5, as they include the surface impact of the windstorms that reach the European continent and they are the input of the impact metric used in this article (section 3.2). The climatologies of the 10 m wind speed are also qualitatively similar in ERA5 and GloSea6 (Fig. 2(c),(f)). 10-m wind speeds are typically larger around coastlines and towards northwest Europe. Smaller scale features identifiable in ERA5 are not present in GloSea6 owing



to the lower horizontal resolution, for example around coastal and mountainous regions (Fig. 7(i)). ERA5 retains information from its native high-resolution grid despite having been regridded to the GloSea6 resolution. Nevertheless, it is apparent in Figure 2 that biases in GloSea6 for the storm track and near surface wind speeds are generally small.

## 4.2  Impact metric

The SSI, and the method to convert from wind speed to wind gust SSI in GloSea6 (section 3.2.1), is now assessed. Distributions

of SSI values for cyclone tracks are shown here for several example countries. Only cyclone tracks for which the SSI value is non-zero, i.e. there is an expected impact in the country, are included in Figure 3. Four different SSI distributions are shown for each country. Two are from ERA5 data: one calculated using the ERA5 wind speed and the other the ERA5 wind gust footprints. The other two are from the GloSea6 data: one showing the direct calculation of the SSI using the GloSea6 wind speed footprints and the other the estimated GloSea6 wind gust SSIs obtained by bias correction.

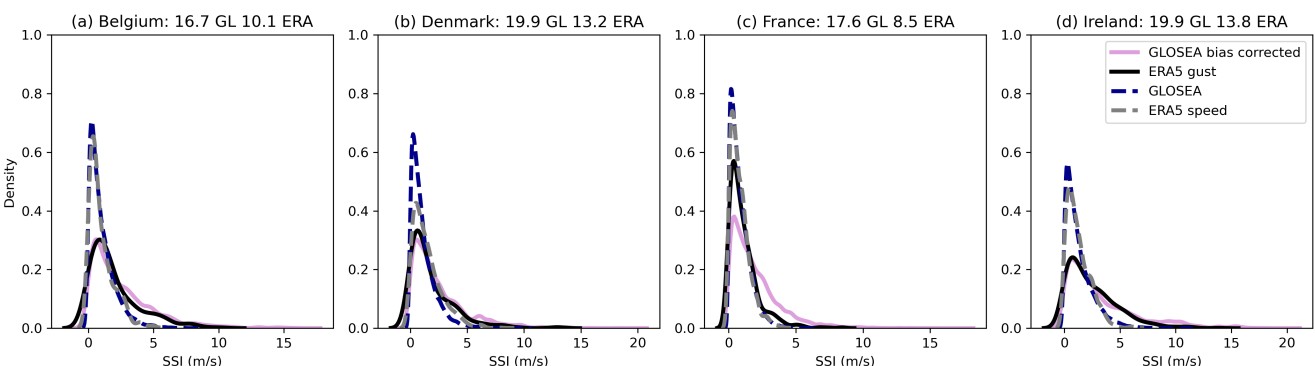

**Figure 3.** Distributions of the storm severity index (SSI) calculated in GloSea6 and ERA5. Wind speed SSIs are shown for ERA5 and GloSea6, together with bias corrected wind speed SSIs from GloSea6 (i.e. GloSea6 wind gust SSIs) and wind gust SSIs from ERA5. The SSI distributions are shown for Belgium (a), Denmark (b), France (c) and Ireland (d) as a representative sample of the countries included in this article. Maximum values of the wind gust SSIs are shown in the panel titles for Glosea6 (GL) and ERA5 (ERA).

In general, the wind speed SSI distributions are similar in ERA5 and GloSea6 for the countries included (comparing blue and grey lines in Figure 3). The similarity between two distributions can be tested using a two-sided Kolmogorov-Smirnov (KS) test. Doing so reveals that the wind speed SSI distributions are indistinguishable for Belgium and France at a 95% confidence level, and Ireland as well if testing for difference at a 99% confidence level. It is also evident in Figure 3 that the bias correction successfully maps GloSea6 wind speed SSIs to wind gust SSI values with similar distributions to the wind gust SSIs calculated

in ERA5 (comparing purple and black lines in Figure 3). As expected, the wind gust SSIs have higher values than wind speed SSIs for both GloSea6 and ERA5, simply because wind gusts have higher values than wind speeds. For certain countries there is a greater difference between the GloSea6 and ERA5 wind speed SSIs (i.e. before bias correction), such as Denmark (Fig. 3(b)), or between the bias corrected GloSea6 SSI and the ERA5 wind gust SSIs, such as France (Fig. 3(c)). The former





is likely due to GloSea6 biases in the wind speed or the number of impactful storms for a country (Denmark is a region of
positive bias in storm track density in GloSea6 (Fig.2(g))). The latter is a result of the quantile mapping approach, which is
built using the climatology of ERA5 SSIs in a country (not just those assigned to a specific storm track), and therefore may
not be representative of the wind storm track defined SSIs presented in Figure 3. The choice to build the quantile map using
the full climatology of SSIs is to ensure sufficient data points are available and the percentiles used in its calculation are well
defined.

To summarise this section, the output from GloSea6 appears appropriate for the aims of this study, i.e. using the UNSEEN
methodology. Additional tests were performed to ensure that the GloSea6 hindcasts were suitable even towards the end of their
evolution (when the model may have drifted towards its own climatology and further from the real atmosphere). Separating
the forecasts into their first and second halves and reproducing Figures 2 and 3 does not result in larger departures between
GloSea6 and ERA5 (though the storm track biases show somewhat different patterns). We therefore conclude that the entire
forecast run is suitable for analysis, which enables the inclusion of a greater number of storm tracks in the following analyses
and therefore the better estimation of extreme wind storm impacts.

## 5   Windstorm statistics

In this section, key characteristics of the identified storm tracks are first compared in ERA5 and GloSea6, before comparing
the most extreme storms to those less extreme in the GloSea6 hindcasts. The cyclone track characteristics that are analysed are:
the storm lifecycle (i.e. how many timesteps in the dataset the system is identified), the mean pressure anomaly of the system
during its lifecycle, the distance travelled (i.e. the sum of the distances between each of the cyclone track points), and the speed
of the system (calculated as the total distance travelled divided by its lifetime). Together, these statistics describe important
features of the tracked cyclones that may influence the SSI. Statistics are shown for any storm track with a point identified in
the North Atlantic/European region, here defined as 20°–90°N, 40°W–60°E.

The track statistics in GloSea6 and ERA5 are shown in Figure 4. There is again good agreement in GloSea6 and ERA5
for each of the track statistics. The distributions of the storm lifecycles, total distance travelled, and their speed are similar in
the two datasets. The strength of the wind storms, evaluated using the pressure anomalies along the storm tracks, are also in
good agreement. This agreement can be seen in the means (green triangles in Figure 4), medians (orange lines), inter-quartile
ranges (rectangle edges) and tenth and ninetieth percentiles (whiskers) of the distributions in GloSea6 and ERA5, with only
marginal differences between the datasets in each case. The track statistics in GloSea6 and ERA5 again highlight the ability of
the GloSea6 system to represent windstorms and its suitability for our purpose.

We can now consider how these storm characteristics differ in the most extreme storms in GloSea6. Extreme storms are
defined here as those that have an SSI value for any of the countries considered that exceeds the maximum identified in
the ERA5 dataset, i.e. the *unprecedented* storms (note that these storms are a subset of those shown in Figure 4). These
unprecedented storms are compared to all the remaining storms in the GloSea6 dataset. The results are shown in Figure 5.
The most impactful storms in GloSea6, based on the impact metric, display, unsurprisingly, different track characteristics. The





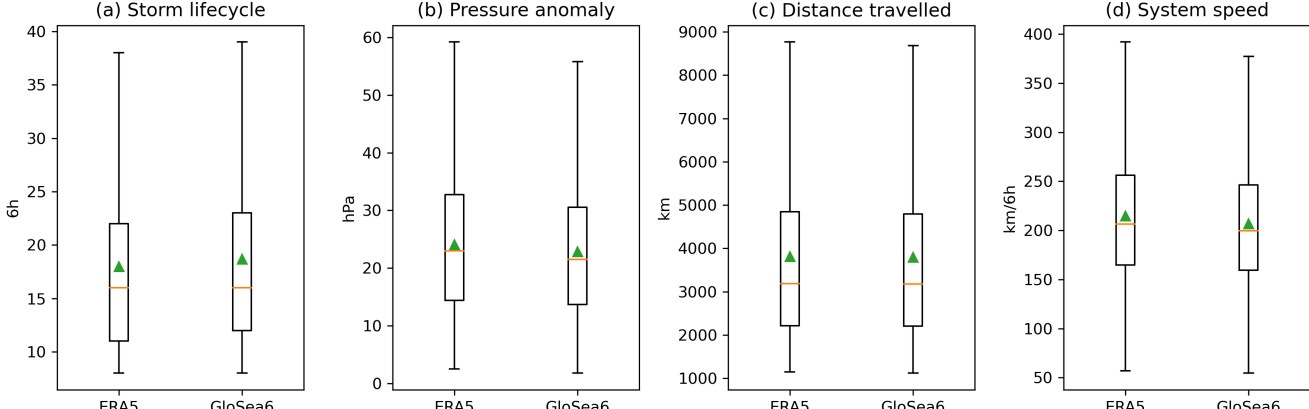

**Figure 4.** Distributions of (a) the number of points constituting each storm's track, (b) the mean pressure anomalies of the cyclones at each of these points, (c) the total distance travelled by the cyclone during their lifetimes, and (d) the average speed at which they move. Distributions are shown for ERA5 and GloSea6 for all cyclones identified in the dataset within the North Atlantic/European domain. Orange lines and green triangles denote the medians and means of the data, respectively.

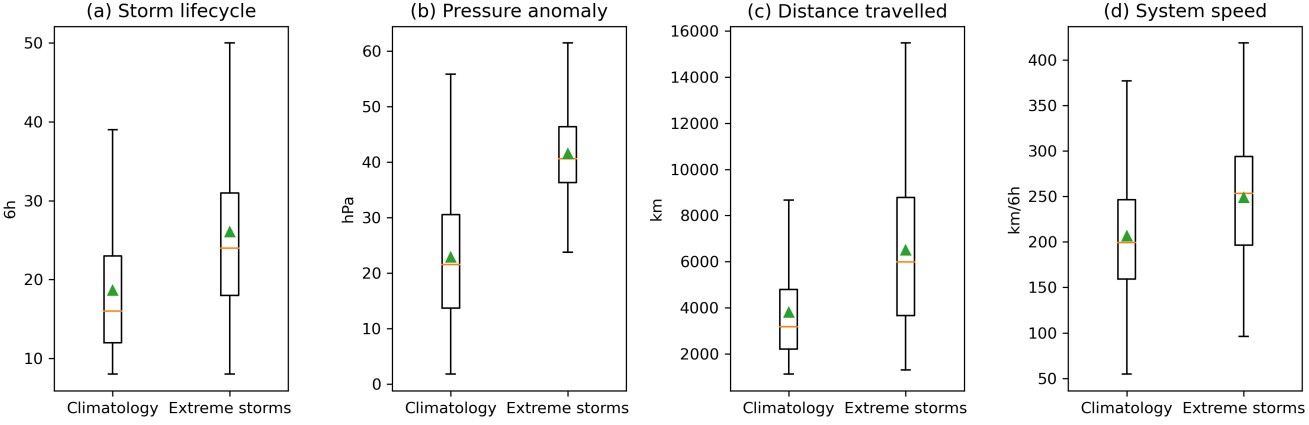

**Figure 5.** As in Figure 4 but comparing the unprecedented storms to the non-extreme storms in GloSea6.

most impactful storms tend to be longer lasting (Fig. 5(a)), have a greater pressure anomaly (Fig. 5(b)), and travel further and at greater speeds (Fig. 5(c,d)). For each track statistic, the median in the extreme storm distribution exceeds the third quartile in the climatology. In the mean, the most impactful storms have a lifecycle that is 36 hours longer than those that are less impactful, have a pressure anomaly over 15 hPa greater, travel around 2000 km further and travel approximately 8 km/h faster (or ~50 km/6h). The distributions are all significantly different if tested using a two-sided KS test. The impactfulness of these unprecedented storms is assessed in the following sections.





## 6   Storm impacts

One of the aims of this paper is to provide another estimate of the impacts from the most extreme wind storms and understand
their potential drivers. First, we calculate storm return periods in the GloSea6 dataset and compare the GloSea6 results to
those obtained from ERA5. The dataset extends the return period of storm impact that we can estimate to 672 years. (This is
compared to around 80 years for the full ERA5 dataset, or the 50 years (1972-2022) used in Figure 6.) This also allows for the
return period of some well-known historical storms to be better estimated using the longer time period dataset.

### 6.1   Storm return periods

The wind gust SSI values, as a function of return period, are shown for ERA5 and GloSea6 in Figure 6 for Austria, France,
Great Britain and the Netherlands. The historical storms in ERA5 are extrapolated using a generalised Pareto distribution
(GDP fit), which is then used to estimate the SSI for higher return periods, which serves as a comparison to the SSI values
for high return periods obtained here using GloSea6. The shape of the SSI-return-period curve is of most interest here, in
particular for the right tail of the distribution that identifies the extreme storms, rather than the specific SSI values represented
in GloSea6 (which will contain some model bias). As such, an additional correction is performed whereby the GloSea6 return
period curves are shifted in the vertical so that the 10-year return period value matches that in ERA5. The shape of the curves
estimated by GloSea, as above, are therefore retained, and by assuming the shape in GloSea is accurate, an estimation of the
potential impacts of the most extreme storms is possible.

For the majority of countries, the SSI values for different return periods generally agree well in ERA5 and GloSea6. This is
not the case for Austria (Fig. 6(a)), where the SSI-return-period curve in GloSea6 lies above that in ERA5 for the lower return
period storms (despite being bias corrected to match ERA5 for the 10-year return period level). There is a large discrepancy
between GloSea6 and ERA5 for Austria because it is a mountainous country, and the relationship between wind speed and
wind gust is less well corrected for by the quantile mapping approach. For mountainous regions, peak wind gusts can reach
very high values but sustained wind speeds are typically much lower, and so a high wind gust value may not necessarily
respond to a high wind speed value (as is typically the case for other regions), which renders the quantile mapping technique
used here less effective. As the majority of countries considered here are not mountainous they do not suffer from a similar
issue. This is evidenced by the more closely matching SSI-return-period curves for France, Great Britain and the Netherlands,
where for the bulk of the distribution the shape of SSI return period curves agree well in ERA5 and GloSea6. In France
(Fig. 6(b)), the GloSea6 SSI values match those in ERA5 for even the most extreme storms, adding confidence to the estimates
of windstorm impacts currently being used in the insurance industry. For higher return period storms in Great Britain (Fig. 6(c))
and the Netherlands (Fig. 6(d)), GloSea6 SSIs are lower than the extrapolation of ERA5 and outside of the GPD fit uncertainty
estimate. This is due to an apparent flattening of the SSI curve with increased return period. The bias correction is converting
the GloSea6 SSI values to a distribution with a flattening tail as the tail also begins to flatten in the ERA5 wind gust SSI
distribution (the GPD fit also flattens off, though not as quickly). The very extreme storms that may have exceptional SSI

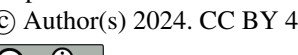



**Figure 6.** SSI curve against return period (note the x-axis is on a log scale), for the GloSea6 event set (orange) and ERA5 event set (black) for (a) Austria, (b) France, (c) Great Britain and (d) Netherlands. SSI values for ERA5 are here calculated using the hourly 10 m wind gust since previous post-processing at 0.25 x 0.25 degree resolution as follows. Event dates in the ERA5 event catalogue are extracted for storms between 1972-2022 and the SSI is calculated based on the 3-second peak gust using a 20 m/s threshold using equation (1). A Generalized Pareto Distribution (GPD) is fitted to the reference catalogue following the Peaks Over Threshold (POT) method. The median fit (black solid lines) and its corresponding uncertainty (black dashed lines) are shown. The uncertainty of the GPD fit results from the selection of the threshold and parameters of the fit. This is reported to 95% confidence intervals. The thresholds for the POT method are selected based on the 85-95th percentiles of SSI values. For GloSea6, SSI values for the full event set are shown in orange crosses, while the orange lines show 20 samples of these, each of 50 randomly selected seasons. An additional bias correction is included here whereby the GloSea6 curve is shifted vertically so that the SSI value at a return period of 10 years is equal to that in ERA5.





values may also not be present in the GloSea6 simulations as they are intialised every season from climatological values and
       are therefore somewhat constrained to the past atmospheric evolution.

       One of the strongest storms in history in Great Britain is storm Daria (with an SSI value around 13.0 it is one of the rightmost
       black marks in Fig. 6(c)), which impacted the UK in January 1990 and caused more than £7 billion of insured losses (as if 2023
       values). The flatness of the GloSea6 SSI curve suggests that the most extreme storms in that dataset have similar SSI values and
that there is considerable uncertainty in their return period. For a severe storm such as Daria, datasets that cover longer periods
       than current historical records are needed to more accurately estimate the return period. This kind of information, comparing
       the most impactful windstorms from recent memory and better constraining their return period, can be used in the insurance
       sector for comparison with catastrophe model output, and ultimately better estimations of the total insured losses that would
       need to be paid out in the event of a very high return period storm. This can be done for each country individually, and for
many of the other well known historical storms, to reduce the uncertainty on the impacts that could be expected from a storm
       that may occur at anytime in the future.

## 6.2   Influence of storm clustering

       The serial clustering of windstorms, which is to a large extent driven by the large-scale circulation over the North Atlantic
       (Mailier et al., 2006; Priestley et al., 2017), affects wind storm numbers and loss totals for Europe (Priestley et al., 2018).
This relationship between the serial clustering of cyclones and windstorm impacts is now quantified in the GloSea6 hindcasts
       and ERA5. To do this, a counter factual dataset is produced with any (non random) clustered periods removed. Clustering is
       removed by randomly assigning each storm to a specific date within a season. This randomly generated dataset will not contain
       the clustering frequency characteristic of GloSea6, i.e. the dynamical clustering of the atmosphere, and any clustering that does
       occur will be due to storms being randomly assigned together. Comparing the AEP in the GloSea6 hindcasts to the random
dataset therefore quantifies the impact of cyclone clustering on seasonal losses. If, on average, the true AEP is larger than the
       random AEP, then cyclone clustering acts to increase seasonal losses. The ratio of the AEPs (true over random) are shown in
       Figure 7 as a function of storm return period, for several example countries.

       The ratio (AEP/AEP_random) increases with increasing storm return period for all countries considered (Fig. 7). This means
       that the most extreme seasonal losses caused by windstorms are more impacted by cyclone clustering than more commonly
observed lower loss values. For these example countries, the presence of cyclone clustering makes losses higher for storms with
       a return period greater than approximately 2 years. Seasonal loss totals increase by approximately 10–20% during clustered
       periods for storms with return periods between 10 and 100 years. The dashed lines in Figure 7 show 10 samples of 100 seasons,
       to estimate the variability in the impact of clustering on seasonal loss totals. There is some variability depending on the seasons
       included (the dashed lines diverge) but generally the ratio is above one, indicating increased losses due to clustering, for
seasonal loss totals with return periods between 10 and 100 years. This offers (re)insurance companies a potential predictor of
       seasonal insured losses that can help inform their decisions based on expected loss totals.







**Figure 7.** The annual exceedance probability (AEP) ratio as a function of storm return period in GloSea6 in (a) Belgium, (b) Germany, (c) Denmark and (d) Great Britain. The ratio is shown between the true AEP in GloSea6 and in a dataset constructed with cyclone clustering artificially removed. The solid line shows storms from all of the seasons combined, whereas the dashed lines show 10 samples of the dataset consisting of 100 seasons.



## 7 Unprecedented windstorms

The UNSEEN approach is designed to inform on events that could plausibly occur at anytime in the current climate, but have not been observed because of the natural internal variability of the climate system and the relatively short observational record. Here, we investigate whether these so-called unprecedented windstorms are found in the GloSea6 dataset for each of the countries considered. The likelihood of an unprecedented windstorm occurring in each country is then calculated followed by an estimation of the influence of the NAO on seasonal windstorm impacts and the likelihood of unprecedented windstorms.

The estimated impact of all of the tracked windstorms (the wind gust SSI values) are shown for GloSea6 and ERA5 for each country of interest in Figure 8 (note that this is the same data as displayed in Figure 3 but for all countries and presented in a way that allows for better visualisation of the extremes). The maximal SSI value of all the storm tracks in each dataset for each country is also highlighted in the panel title. The bulk of the distributions, i.e. the medians and inter-quartile ranges, of the wind gust SSIs are similar for ERA5 and GloSea6 for the countries considered. The upper quartiles in GloSea6 tend to be slightly higher than ERA5 in most cases. This may reflect biases in the GloSea6 wind speeds (which tend to be slightly positive for the countries considered), a feature produced in the bias correction, or simply the greater sample size including more impactful storms with higher SSIs. Nevertheless, the broad similarities between GloSea6 and ERA5 suggests that the bias correction procedure is successfully converting GloSea wind speed SSIs to values representative of wind gust SSIs and are hence useful for the evaluation and development of catastrophe models used at (re)insurance companies. Unprecedented windstorm impacts are found in the GloSea6 hindcasts for nearly all of the countries, and therefore these storms should be expected to cause unprecedented damage. The extent to which the GloSea6 storm impacts exceed the maximum in ERA5 varies by country. For the majority of countries, the most impactful storm in GloSea6 has an impact approximately 1.5 times stronger than the most impactful storm in ERA5 (e.g. Belgium Fig. 8). These storms would result in considerable increases in insured losses.

The number of unprecedented windstorms, and hence the likelihood that one should occur, depends on the country. The likelihood of an unprecedented storm is calculated by counting all the unprecedented storms for a country and comparing to the total number of storms that have any impact for that country in the GloSea6 hindcasts. Counts range from zero for Austria and Finland to more than 600 for Great Britain over the 672 year event set (green circles in Fig. 8). More than 100 storms that would be expected to produce impacts more severe than any in history are simulated in GloSea6 for the majority of countries. Comparing the counts of unprecedented wind storms to the total number of storms impacting a country, it is found that unprecedented windstorm impacts are found, in general, for between 0.5 and 1 percent of all GloSea storms impacting countries within Europe (grey squares in Fig. 8). In other words, there is between a 0.5 and 1% chance that a storm that impacts a country will be unprecedented. Generally, unprecedented windstorms are most frequent for countries in the jet exit region (e.g. Great Britain, Ireland and France). This may reflect the slight GloSea6 biases in this region (Fig. 2). We have less confidence in the results for mountainous countries where the wind speed - wind gust relationship is more variable and the bias correction less effective, such as Austria (not shown).





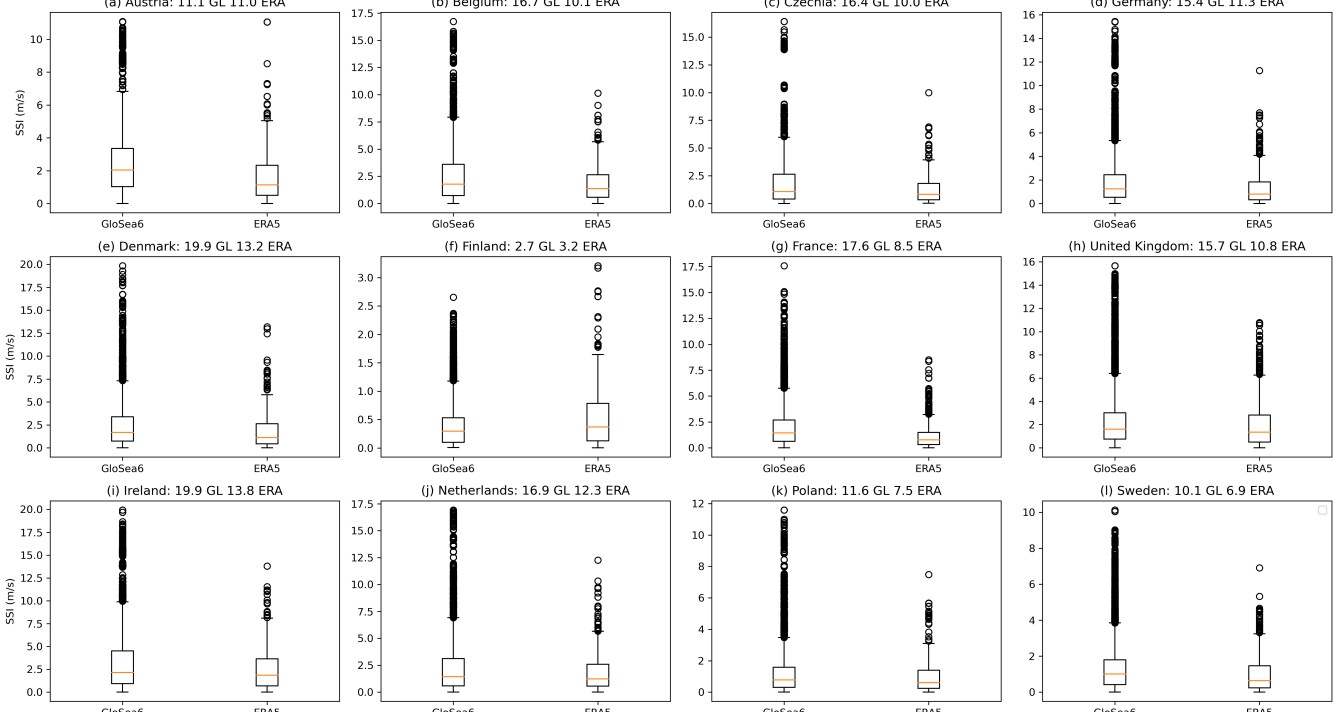

**Figure 8.** Distributions of the wind gust SSI for each country included in the analysis for GloSea6 and ERA5. The values shown in the panel titles are the maximum SSI value in GloSea6 (GL) and ERA5 (ERA).

## 7.1 Influence of the North Atlantic Oscillation

Previous studies have shown that certain characteristics of the large-scale circulation, such as the NAO, influence the frequency and severity of windstorms, and their impacts, reaching Europe (Donat et al., 2010; Dawkins et al., 2016). Here, we assess if the NOA influences both seasonal loss totals and the likelihood of occurrence for the unprecedented windstorms. There is potential for such large-scale circulation patterns to act as predictors of windstorm loss totals and unprecedented windstorms, as they may be better predicted at seasonal timescales than the windstorms themselves.

The NAO describes the large-scale pressure pattern over the North Atlantic region, which is related to storminess over Europe (Alexander et al., 2005; Matulla et al., 2008; Feser et al., 2015). In its positive phase, the pressure gradient between southern and northern midlatitudes is increased, the jet stream is strengthened, and the frequency and intensity of extratropical cyclones reaching northwest Europe is elevated. Severe storms can still occur in the negative phase of the NAO (Pinto et al., 2009). Previous studies have documented the link between the NAO and storm impacts over Europe (Degenhardt et al., 2023;

Priestley et al., 2023). Here, we consider two ways the NAO relates to storm impact. First, we quantify how the winter mean NAO value relates to total extended winter storm losses in both GloSea6 and ERA5. Then, the probability that an unprecedented





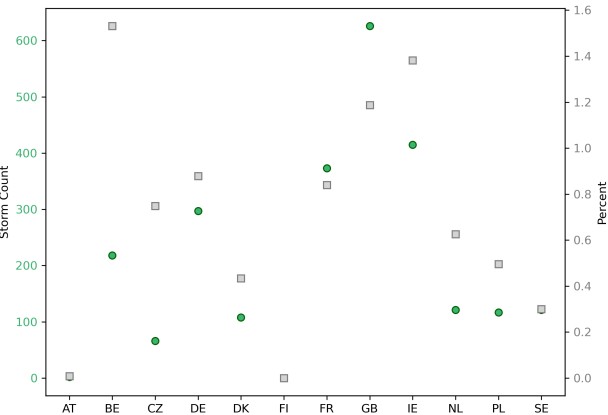

**Figure 9.** The number of unprecedented storms in the GloSea6 dataset for each country (green circles, left y-axis). The percent of all of the storms impacting a country that are unprecedented is also shown (grey squares, right y-axis).

windstorm occurs in the GloSea6 hindcasts is compared during opposite phases of the NAO. The correlation between the total losses in a season (i.e. the AEP) and the winter mean NAO value is shown in Figure 10(a) for GloSea6 and ERA5.

In ERA5, the AEP is correlated with the mean NAO value: when the NAO index is more positive the total losses are increased
(Fig. 10a). Correlations are slightly weaker for countries further east, e.g. Poland and Sweden, where the influence of the NAO is typically weaker. The correlation is also weaker in France. France sits at the midpoint of the dipole of typical NAO impacts, and the anomalies in relevant variables remain relatively consistent for oppositely signed NAO values and the correlation with the AEP is lower. There also exist correlations between the AEP and NAO index in the GloSea6 hindcasts. The correlations are weaker in GloSea6 compared to ERA5, though they are statistically significant.

The change in likelihood of an unprecedented event occurring in GloSea6, given the state of the NAO, is shown in Figure 10(b). The probability that an unprecedented event occurs approximately doubles on occasions when the NAO index is anomalously high (above its ninetieth percentile), though this is country dependent and the change in probability varies from 1.5 times more likely to nearly 3 times. The probability of an unprecedented storm generally depends more strongly on the state of the NAO for countries further west, such as Great Britain, where the probability of an impactful storm (i.e. a storm
with a non-zero SSI in a country) being unprecedented goes from 1 in 83 to around 1 in 30, and Ireland, where the probability changes from 1 in 71 to 1 in 29. Conversely, on occasions when the NAO index is strongly negative (below its tenth percentile) the likelihood of an unprecedented events drops to roughly half its typical value for most of the countries. Knowledge of the mean state of the NAO in a hindcast run could therefore be useful in preparedness for extreme windstorm impacts. If this can be well forecast at longer lead times, and better than the prediction of individual windstorms, the skill horizon of aggregated
extreme windstorm impacts could be extended.



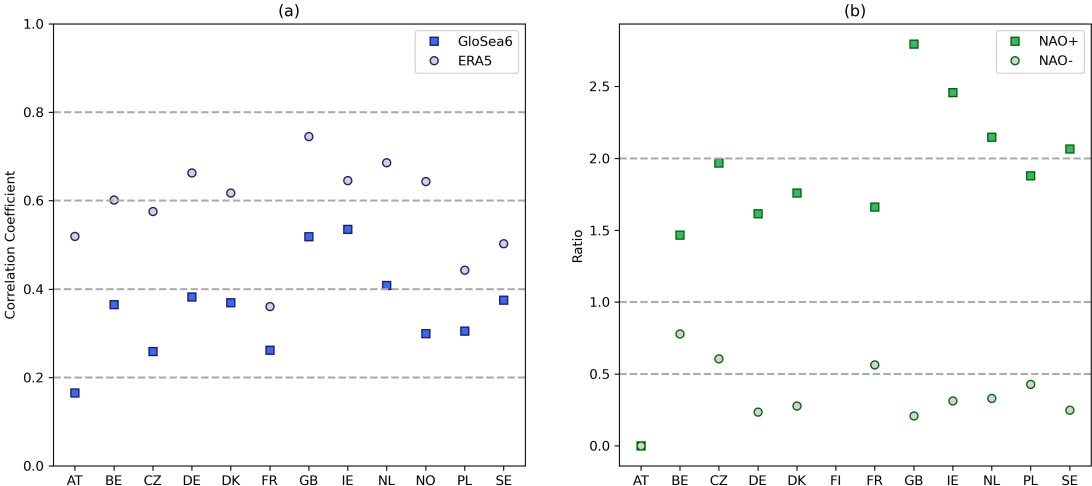

**Figure 10.** (a) Correlations between the annual exceedance probability (AEP) and the seasonal mean NAO values for each country considered in the GloSea6 and ERA5 datasets. (b) The change in likelihood in GloSea6 of occurrence of unprecedented windstorm impacts given a strongly positive (NAO+) or strongly negative (NAO-) NAO index, defined as above the 90th percentile of NAO values or below the 10th, respectively. Dashed lines in (b) represent no changes in likelihood (1.0), a doubling in likelihood (2.0) and halving in likelihood (0.5).

## 8 Conclusions

Output from the Met Office's seasonal forecast system (GloSea6) was used here to create nearly 700 years of extended winter-time extratropical cyclone tracks. Unprecedented impacts from these cyclones, caused by their associated wind footprints, are quantified using the UNSEEN methodology (Thompson et al., 2017; Osinski et al., 2016) and a commonly used storm severity
index (SSI, Klawa and Ulbrich, 2003). Applying the UNSEEN methodology to windstorms impacts allows the first quantification of the likelihoods of more extreme windstorms than ever observed to be experienced for several European countries. The probability of an unprecedented storm impacting a country is generally between 0.5% and 1.6%, for the northern/central European countries considered herein. The GloSea seasonal forecast system is known to represent windstorms with reasonable accuracy (Degenhardt et al., 2023), and additional analyses presented here confirmed its fidelity in windstorm representation.
A better understanding of extreme windstorm impact is of key importance to society, as they are among the most damaging and the most costly natural hazard impacting Europe (Munich Re, 2015) and are expected to become more severe in the future climate (Schwierz et al., 2010; Priestley and Catto, 2022).

The GloSea6 seasonal forecast system was shown to represent extratropical cyclones and a metric of their impact well enough to follow the UNSEEN approach (Thompson et al., 2017) for estimating extremes. Several aspects of windstorms
were compared in GloSea6 and ERA5 to ensure this. Firstly, the number of storms identified within a season, and their mean intensities were shown to be similar in the two datasets. Then, looking at the individual cyclone tracks, the storm lifetimes,



distance travelled and average speeds, together with the pressure anomalies along their tracks, were shown to be comparable in GloSea6 and the ERA5 reanalysis. The distributions of the SSI values computed from the track footprints were again closely matched in the hindcasts and reanalysis. Considering all of this, the GloSea6 output was deemed suitable for creating around 700 years worth of cyclone tracks and studying their impacts. Similar approaches were followed in Lockwood et al. (2022) and Osinski et al. (2016) to create datasets containing a large sample of windstorms, with this work adding adding to this body of research. The model used here, as well as the method to convert from wind speed to wind gust SSIs, distinguishes the presented results from these previous studies. Additionally, the analyses of results with respect to the historically most extreme storm, i.e. focusing on unprecedented windstorms, as well as performing all analyses at a country level, are novel approaches to estimate extreme European windstorm impacts.

A quantile-mapping bias correction approach was shown to successfully convert values of a wind-speed SSI to those representative of a wind-gust SSI. The latter is most commonly used in the insurance industry, as is the aggregation of windstorm impacts to the country level as is done here, and hence the results presented are directly comparable with catastrophe model output, and immediately available for their development and evaluation. Datasets with only wind speed output available and not wind gusts, such as GloSea6 and other seasonal forecast systems as well as climate model simulations, can therefore be used in comparison with catastrophe model output, giving an opportunity for the creation of even larger datasets of windstorms and their impacts. The impacts of extreme storms, estimated with the bias corrected SSI, were quantified and compared to the reanalysis. Hundreds of storms more impactful than any known in the historical period, so called unprecedented windstorms, were found for the majority of countries considered, with between 100 and 600 unprecedented windstorms identified for most countries. This reflects a probability of a storm being unprecedented between 0.5% and 1.6%. The unprecedented storms typically have longer life cycles, stronger pressure anomalies, travel further, and do so at higher speeds.

The impact of higher return period storms can be estimated with the new dataset, which enables the return periods of historical well-known storms to be better constrained. The phase of the North Atlantic Oscillation (NAO) is related to extreme windstorm impacts: seasonal SSI totals are positively correlated with the seasonal mean NAO index and the likelihood of an unprecedented storm is approximately doubled when the NAO index is strongly positive (and roughly halved when the NAO index is strongly negative). Seasonal loss totals are also increased during periods of cyclone clustering (in agreement with Priestley et al. (2018)), particularly for storms with return periods greater than 10 years. These relationships to the large-scale atmospheric circulation offer potential routes of extended predictability for windstorm impacts, with the NAO reasonably well predicted many months in advance (Scaife et al., 2014).

The Climate Data Store archive contains many more thousands of years worth of data from different seasonal forecast models and their ensemble members. Further work analysing these data would improve the quantification of unprecedented windstorms and their impacts presented here, thus improving confidence in the estimates of the potential impacts of the most extreme present-day windstorms and hence, for insurance and reinsurance companies, estimate of the amount of capital that must be held to be able to pay the claims that would arise from such a storm. This is crucial knowledge for the insurance sector.



*Code and data availability.*   ERA5 and GloSea6 data are freely available from the Copernicus Climate Data Store (CDS, 2023). The cyclone tracking code is available from the authors upon request.

*Author contributions.*   JWM wrote the manuscript and performed the majority of the analyses. CHJN performed the return period and storm clustering analyses. All authors contributed to the design of the study as well as providing feedback and comments on previous drafts of the manuscript.

*Competing interests.*   The authors declare they have no competing interests.

*Acknowledgements.*   The authors wish to thanks Kevin Hodges for his support setting up and running the cyclone tracking algorithm. This work used JASMIN, the UK's collaborative data analysis environment (https://jasmin.ac.uk, Lawrence et al., 2013). Guy Carpenter & Company Limited funded this research.



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
