# Peer review of "Using seasonal forecasts to enhance our understanding of extreme European windstorm impacts"

_EGUsphere, 2024_

## Author Comment (AC1)

In this response to reviewer document, the reviewers' comments appear in black and our author responses appear in blue. As several of the reviewers shared similar concerns we begin with a response addressing these over-arching comments.

The main concerns expressed by the reviewers are summarised as:

1. The work is not novel or does not contain sufficient new science to warrant publication.

2. The implications and significance of the presented work are overstated.

3. Prior literature on the topic is not discussed properly or missed altogether.

Our response to each of these points is provided below.

*1. The work is not novel or does not contain sufficient new understanding to warrant publication.*

The reviewers suggest that the work presented does not go sufficiently beyond previous literature on the topic of using models to estimate windstorm impacts. Several articles are listed that have used similar approaches to do this. We briefly explain how the presented research goes beyond or differs from the articles mentioned by the reviewers below. It is worth noting however that simply because previous publications have estimated a particular quantity, in this case extreme windstorm impacts, it does not mean that new estimates of that quantity should not be welcomed and published alongside the prior research, particularly when the estimates of that quantity contain uncertainty, as estimates of extreme windstorm impacts do.

**Walz and Leckebusch (2019):** use a seasonal forecast model and natural catastrophe damage model to estimate windstorm impact. This is indeed broadly the same approach used here. However, this study only estimates impacts for Germany, the UK, France and Spain. Our analysis therefore includes estimates for eleven additional countries. Providing estimates of windstorm impacts using a seasonal forecast model for these additional countries is done here for the first time (to the best of our knowledge). It is important to understand the potential windstorm impact for these countries, particularly for the insurance industry, which provides insurance to these countries, as well as for communities living in these regions that wish to understand extreme windstorm impacts.

In addition, the seasonal forecast system used in Walz and Leckebusch (2019) is different to that included here. Befort et al. (2018) show that there are differences in skill for windstorm/ extratropical cyclone prediction between seasonal forecast systems, so an analysis of windstorm impacts using a different seasonal forecast model should provide valuable information on the uncertainty in the estimates obtained in Walz and Leckebusch (2019), and be useful for the research and insurance communities. Indeed, Walz and Leckebusch (2019) find that, for the seasonal forecast model they use, the reanalysis based loss estimate can lie outside the spread of estimates from the model (for the most stormy winters) and that the mean loss predicted by the model can differ from that in the reanalysis for Germany. Repeating similar analysis to that in Walz and Leckebush (2019) using a different model can help better understand if loss estimates obtained from seasonal forecast models are sensitive to the choice of model.

**Osinki et al. (2016):** use medium range forecasts from an EPS to estimate windstorm impacts. 15-day integrations of the EPS are considered during a 10-year period and, as such, windstorms are quite closely constrained to the climate observed during this 10-year period. A method to detect storms produced by the EPS that have no common counterpart in the reanalysis, and therefore potentially may be more extreme than any observed, is used. The method is based on the timing and location of the storms with storms that are unique to the EPS identified as those more than 1500 km from a contemporaneous storm in the reanalysis. Whilst these storms may be purely EPS driven, they are still constrained closely to the climate (particularly the SSTs) of the particular climate state that happened in reality during the time period used, and may miss extreme windstorms that could have developed if the climate had evolved along a different path. The 215-day integrations used here, while still somewhat constrained to the true climate pathway (the SSTs are initialised to the best estimate of the September SSTs in reality) may evolve into situations further removed from the true evolution of the climate system but still following an evolution that is

plausible. That is, assuming we trust the GloSea6 climate evolution is realistic enough, which is an assumption (that is tested) here. Osinki et al. (2016) also aggregate SSI to storm level and compare values found in the EPS and the reanalysis. Our study differs in its focus on impacts at the country level, where potentially different biases may be identified and different information is provided.

**Berfort et al. (2019) and Degenhardt et al. (2023):** these studies both assess windstorm representation in seasonal forecasts and show they provide some skill in the prediction of European storminess, thus providing motivation for using seasonal forecasts to study windstorm impacts as is done here. Befort et al. (2019) also find differences between seasonal forecast systems for features of cyclones and windstorms relevant for the results presented here, such as track densities, relation to reanalysis and dependence on the large scale. This motivates the study of windstorm impacts in multiple different seasonal forecast models, such an analysis is done using GloSea6 from the Met Office for the first time here and complements the studies mentioned above using models from the ECMWF.

In summary, the novel aspects of the presented manuscript, that go beyond any results published in the literature (to the best of our knowledge), and in particular those listed by the reviewers, are a result of:

- The focus on windstorm impact at a country level for 14 countries within Europe. This goes beyond previous studies considering single or few countries or studies that focus on storm impact without separating it into country specific impacts.

- The analysis of windstorm impacts in the GloSea6 seasonal forecast system. One other study (Walz and Leckebusch 2019) has used a different seasonal forecast system (run to seasonal time scales) to investigate windstorm impacts in seasonal forecasts (others have used climate or medium-range weather forecast models). An investigation using another model is undoubtedly useful for the scientific and insurance communities to better understand if estimates of high return period storm impacts are model dependent (especially as the studies listed above show skill is model dependent).

- The quantification of unprecedented windstorm likelihood. Other studies considering windstorm impacts have not quantified the likelihood of occurrence of a windstorm more extreme than any observed (for individual countries using this seasonal forecast system). This could of course have been calculated using the data produced in previous studies, but no such calculations were presented and the focus on unprecedented windstorms is unique to this manuscript.

We appreciate that these do not constitute a significant advancement in our understanding of windstorm impacts. However, we believe there is value in reproducing previous results with slightly different methods and datasets in order to confirm these previous results or give further estimates of their uncertainty. In a revised manuscript we will make it explicitly clear what is new here and how it builds upon and fits within the state-of-the-art literature.

*2. The implications and significance of the presented work are overstated.*

It was not our intention to overstate what the results in the submitted manuscript represent. As no specific examples of where we have overstated the implications or significance of the results were given, it is hard to respond to this comment. In a revised manuscript we will take extra care to ensure any stated implications or implied significance are robustly backed up by the analysis undertaken.

*3. Prior literature on the topic is not discussed properly or missed altogether.*

We disagree with review statements such as "It is clear that the authors have missed a lot of relevant literature on the topic" or "failure to take a holistic view of existing work risks undermining

this": only two of the articles listed by reviewers were not referenced in the original manuscript and the introduction (of approximately 1500 words) of the article was written to summarise the state-of-the-art understanding of the topic. The work by Walz and Leckebusch (2019), which is very relevant for the results presented here, unfortunately did not appear in our initial search for relevant literature on the topic when writing the literature review, so a discussion of the work will be included in a revised manuscript.

As these high level concerns regarding the publication of the manuscript remain at this stage, in the following we address the comments relating to overall quality of the manuscript and validity of the methods. The more specific points related to clarity, and technical or typographical changes, we leave for later in the review process.

Reviewer 1

Summary

The authors use ensemble predictions to enlarge the data base to assess extreme storms over Europe. The aim is to better constrain extreme events which is highly relevant for society given their strong economic impact. The authors state that the data set is suitable to be used in this respect and confirmed the relation ship of the NAO to influence European storms. They also highlight that serial clustering of windstorms impacts the impact on the country level.

General

The paper provides some interesting insights in using forecast data to better constrain extreme windstorm impacts. Still the paper suffers from some problems in the method section and structural issues concerning the evaluation and result sections which need to be solve prior to possible publication. Moreover, I got the feeling that something with the bias correction approach is wrong or at least not well explained. Thus, I recommend major revisions.

We thank the reviewer for offering their time to review the manuscript. Please see above for our response to the first round of reviews and below for responses to your comments relating to the methods/novelty of the manuscript below.

Comments

Title: Given in content of the manuscript I find the title a bit misleading I rather think the aim of the study is to use seasonal forecast to better constrain and assess extreme windstorm impacts. A mechanistic understanding is not presented in the manuscript.

Abstract

L2-3: I suggest removing the second sentence of the abstract as this is just a repetition of the introduction and fits better there.

L 3: Start the third sentence with "We use nearly…"

L11: Please remove "strongly" before "increase".

L12 two points.

L14: I am not aware of any publication which shows predictability of the NAO several months ahead. The authors also do not present any publication on this so avoid such a statement in the abstract and give references in the main text if such publications exist.

Scaife et al. (2014) state that "The winter NAO can be skilfully predicted months ahead". This will be added in the main text.

L30-34: The authors give the impression that one needs to have several hundred years of data to estimate return periods of rare events, but in principle extreme value theory is developed to this for shorter data sets. It is its strength to use say 30 years of observations and estimate a 100-year return period. Clearly the uncertainty will be high, but it is possible. Please be more balanced here and say that longer time series will lead to a reduction of uncertainty.

L48 and elsewhere in the manuscript: Please change methodology to method.

L66 please change "creation" to "generation".

L67: The authors noticed a certain problem of the data set used. Unfortunately, they did not assess whether a potential drift may impact their results. They can do this, and I think such an analysis would be beneficial for the publication.

We did check that the SSI distributions were similar in the early and late part of the forecasts (they were). This will be emphasised in a revised manuscript.

L90: I checked the Degenhardt publication, and the seasonal predictability of windstorms is not overwhelming and over land not existent, so please clarify.

L114, L122 and L123: I am puzzled about that the data is somehow downscaled without saying with which method (dynamically with a regional climate model simulation or statically). I guess that the authors misuse the word downscaled here as they would also say from which resolution they downscale. So please clarify.

The data we used is not downscaled in anyway. These sentences refer to data being downloaded not downscaled.

Data section: Another issue is that ERA5 is not well introduced and it remains unclear at which resolution ERA5 is used. In section 2 it sounds like that it is interpolated to the resolution of GloSea6 whereas in Fig6 the authors use it in 0.25 degree resolution. Why is this changed (it is maybe a bit unfair to sue ERA5 in higher resolution).

L145 do you use wind speed or wind gust?

L158: The sentence is unclear. Do you only calculate one SSI value per track and what happens if the cyclone track travels over several countries? In the SSI definition the authors use an area around the cyclone center, how is this split if the area covers more than just one country. To me the SSI calculation is rather unclear.

L171-184: Somehow I get the impression that the authors are not aware of the fact that the wind gust provided in ERA5 is a diagnostic variable which is mainly based on surface wind speed. So I expect that if the authors following the 2% criteria of Klawa and Ulbrich (2003) and then check in the wind speed that they find that the threshold is in the range of 96th to 98th. In principle it must be 98th percentile and as gust and windspeed are strongly linked it must be also 98th percentile for wind speed.

Table 1: you show wind speed not gust correct?

L185-199: The description of the bias correction is unclear. It sounds like that the authors apply the quantile mapping method between ERA5 wind speed and ERA5 wind gust and use the coefficients

in GloSea6. In this case no bias of GloSea6 is corrected. The authors only made a scaling to obtain from wind speed gust-based SSI, but then name it scaling. IF you would like to do a bias correction you need to estimate coefficients (of regularly spaced quantile intervals) between GloSea6 wind speed and ERA5 wind speed and then scale it to ERA5 gust.

You are correct that we did not directly bias correct GloSea6. We referred to the conversion from windspeed to wind gust as bias correction for both ERA5 and GloSea6 for simplicity, but we will rename it as a scaling, as suggested, in a revised manuscript.

L200: The definition of NAO is not fully clear; do you normalize the mslp difference or the northern and southern center separately. Please clarify which definition you use, is it the one of Hurrell (1995)?

L206-7: The intro sentence is not needed.

Section 4: Here I recommend enhancing the evaluation to the question of whether the 215-day predictions drift and whether such a drift impact the results presented. For this I suggest to analyses the first 1/3 of the 215 days and contrast the results with the last 1/3 of the 215 days.  Do the biases, shown in Figure 2, change? What about Fig 5, do we see the increases if we restrict to the first 1/3 and the last 1/3?

L215: I find the track changes quiet string, the storm track is more zonal in GloSea6 compared to ERA5, Changes are in the order of 10%. Please apply a significance test in Fig 2 so the reader concentrates on the important biases.

L234: remove comma after (section 3.2.1)

L235 Remove "here".

Overall, I find a lot of bracket in the manuscript, maybe the authors can remove a couple or avoid them by making real sentences. It would increase readability.

Section 5: Form me this is more an evaluation section so merge it with section 4 and make it a subsection 4.3

278: Please remove "here"

Fig 4d and 5 d Please change the unit to km/h

L284: I do not see a 36 h increase it looks like a 6 h increase in Fig 5a.

L285: according to Fig 5b I would say it is rather a 20hPa increase of the pressure anomaly.

L289-293: This is an introduction to the results part but only section 6.1 is introduced, please do this with 6.2 and 6.3 (I suggest renaming section 7 to subsection 6.3).

L299-301: I am puzzled here, why do the authors need to apply an additional "bias" correction if the used a bias correction already. To me this is a sign that something in the proposed method is wrong or at least not well explained.

This is indeed a specific characteristic of how we performed the bias correction, and perhaps could have been explained better. The initial bias correction, which uses the ERA5 quantile map on the GloSea6 output, is not intended to match the seasonal forecast data to the reanalysis perfectly (though this could be done easily by constructing Glosea6 —> ERA5 quantile maps), as this would prohibit the identification of extreme windspeeds beyond the reanalysis maximum. The additional bias correction (which is only used to make the data shown in Figure 6) is done to account for situations when the ERA5 quantile map is less effective in a particular country. For most countries this additional "bias correction" was small. It had the biggest impact for Switzerland and Norway

(and to a lesser extent Germany). We were interested in what the GloSea6 event set was telling us about the extreme events in the tail above the length of the reanalysis period but believed the reanalysis gives a more realistic representation of storm return period up to around 50 years, so the additional bias correction was done to reduce any bias in GloSea6 up to this point (which potentially remained after the initial bias correction using the ERA5-quantile maps). In a revised manuscript, we will make the method followed clear and give our motivation behind the choices.

L306-311: The different behavior in mountain regions is again weird as a good bias correction on the region level show be able to correct for this issue.

See response to above comment. Mountain regions are identified as having different wind speed distributions in ERA5 and Glosea6 (the GloSea6 windspeeds are underestimated) and hence the ERA5 constructed quantile maps are less effective. This will be made clear in a revised manuscript.

Fig. 6 Why do the authors suddenly use ERA5 in a higher resolution? I think this is an unfair comparison.

L322-331: To me the interpretation of Fig 6 c with the single event Daria is unclear. The leveling off in Fig 6 c and also in the other examples looks like a resolution issue, I speculate that SSI is somehow restricted to a certain threshold give the resolution of GloSea6. So to me this looks like more of a limitation of the data set.

Section 7: I suggest renaming it 6.3 and avoid 7.1 as only subsection which make structurally no sense.

L358-371: I am puzzled about the paragraph. I thought the entire procedure suggested enables us to see whether season forecasts can be used to enhance the data base so that unseen extremes can be assessed. But part of the paragraph read like the author doubt their own method and biases are too strong.

It was not our intention for the reader to doubt the method here, we will rephrase in a revised manuscript.

L370. How do you come to the conclusion that the impact is 1.5 time stronger? How is this estimated.

L375: For me this is a strange result and may show that the bias correction has not worked probably. The Authors state that they find in 672 years 600 events which exceed the strongest of ERA5 (which has a 50 year span). This seems to be very unlikely and hints more that there is a substantial bias over GB.

L384: please remove this line.

Fig 8 and Fig 9. Labels are too small, please revise.

Conclusions: I miss to see a discussion of limitations, e.g. the still rather coarse resolution of the data set (though I agree it is better to most of the climate models used). Then there is also the problem of wind gust parameterization also in ERA5 which could be mentioned here.

Gregor C. Leckebusch

The manuscript describes the use of seasonal hindcasts to enhance the understanding of extreme European windstorm impacts. The manuscript claims to go beyond existing literature. Nevertheless, it should be noted that this approach was probably firstly introduced in Walz & Leckebusch (2019) and further related work to this wider topic is published in Walz et al. (2018), e.g. for the link to large-scale modes.

Please see the text at the start of this document for our response regarding the overall novelty of the present manuscript.

Walz and Leckebusch (2019) investigated the feasibility and added value of using the seasonal hindcasts of the ECMWF System 4 as a hazard event set for European winter windstorms damage calculations. The windstorms are identified for every ensemble member and every year by an objective windstorm tracking algorithm. The damages are calculated directly from the obtained wind footprints via the open source natural catastrophe damage model CLIMADA for Germany, the UK, France and Spain and compared to the loss from ERA-Interim. The results show that the ensembles of losses in System 4 nicely capture the inter-annual loss variability of the reanalysis. Due to more than 1,500 years of "virtual reality" windstorm data from the hindcasts, the return levels of extreme losses can be estimated fairly accurately. Based on System 4, the losses in the scale of 1990 (January, February, March and December including the prominent windstorm Daria) represent a 20-year event in Germany whereas they represent a 100-year event for the UK. Thus, a considerably shorter return period compared to return periods calculated from ERA-Interim alone.

Further they investigated the link between the annual losses and large-scale drivers derived from mean-sea-level-pressure (MSLP) data in System 4. They could show that within System 4 there is a significant link between increased loss potentials for strongly positive North Atlantic Oscillation (NAO) phases for Germany and the UK as well as a reduced loss potential for Spain. The link between the other analysed indices is weak bar the East Atlantic (EA) pattern index. Thus, if the NAO in System 4 is correct it can be assumed that the windstorms in System 4 are useable. If this premise is given their study shows that the loss estimates and ultimately the return levels of losses from System 4 can be used in an operational way.

Thanks for pointing to this study that we had missed in our initial review of the literature. A revised manuscript will contain a discussion of this work and how it relates to the analyses presented here. We will include in a revised manuscript a summary of the findings described above and how they complement and/or differ from our findings.

Also, while Klawa and Ulbrich (2003) introduced the notion of the 98th percentile for loss estimates, the SSI was introduced in Leckebusch et al. (2008), as an objective (!) identification and tracking approach to assess the severity per event without arbitray assumption (e.g. for all days of an event the windspeeds in a state or a fixed radius are used). Details on the differences are especially important for the assessment of return periodes or the wider physical characteristic of the event (e.g. related to the SSI footprint as introduced in Leckebusch et al. (2008)).

We agree that the exact definition of SSI is important and should be carefully explained. Our choice of definition for SSI was made to best align with what is used in-house at (re)insurance companies. We will emphasise this in a revised manuscript.

The manuscript also comments on the events deduced in Osinski et al. (2016). Here, it seems the authors assume that the pure event-set events identified and used in Osinski et al. (2016) would be similar to observations: "As such, the windstorms produced are not independent from observations and are somewhat constrained to the climatology of the period (e.g. the SSTs)." While the latter is of course correct (but also applying to the seasonal hindcasts in principle), it should be noted that Osinski et al. (2016) developed especially a method to identify and utilise pure ensemle events to seperate from just similar events as apparent in the reality/observations. Please confer their Fig. 10 for details.

Whilst the method used does identify pure ensemble events, or at least events not accurately predicted by the forecast ensemble members, the cyclones will still be quite closely constrained by the climate (SSTs) of the period. The SSTs during the 10-year period included in their study may not span the range of SSTs that could be realised in the current climate, and therefore very extreme cyclones developing over these SSTs may be missed. This is also somewhat true of course in our study, but the longer time period of the study, and the fact that the model is fully coupled and integrated for 215 days, ought to allow the SSTs to span more of the plausible state space and therefore the potential impact state space of the windstorms. We can add some discussion on these points in a revised manuscript.

These three aspects should be clarified, before publication, idependent from any in-depth review process.

References:

Walz, M.A., M.G. Donat and G.C. Leckebusch, 2018: Large-Scale Drivers and Seasonal Predictability of Extreme Wind Speeds Over the North Atlantic and Europe.

Journal of Geophysical Research – Atmospheres; 123 (20), 11,518-11,535. https://doi.org/10.1029/2017JD027958

Walz, M.A., and G.C. Leckebusch, 2019: Loss potentials based on an ensemble forecast: How likely are winter windstorm losses similar to 1990?

Atmospheric Science Letters, Volume: 20, Issue: 4, Article Number: UNSP e891. https://doi.org/10.1002/asl.891.

Reviewer 2

As previously noted by other commentators the manuscript overstates the significance and novelty of the work. Past works mentioned in previous comments have considered larger datasets or unprecedented storms and it is not clear enough what new contribution is made here.

Please see main response at the top of the document for a clarification of what is new here.

I also feel the merits of seasonal forecasts are overstated in comparison to climate models given the track density biases shown which peak in the most relevant areas for loss modelling.

The biases shown are nonetheless smaller than most climate models. Another important aspect is that the seasonal forecasts are initialised every September and therefore cannot drift too far from the true climate, whereas climate model runs are typically multi-decadal and will drift to the climatology of the model, which may be biased. The benefits of using seasonal forecasts will be better discussed in a revised manuscript.

The climatological period of 1993-2016 is quite short and so although a large number of forecast winters are developed they are based on a narrow window in climatology terms. This is not an insurmountable issue but is due some consideration particularly when examining the effect of the NAO.

This caveat will be discussed in a revised manuscript.

There is some merit in the presentation of the results to be useable by the insurance industry and there is potential for a collaboration to develop a validation dataset which can be more useable than say pure ERA5 but failure to take a holistic view of existing work risks undermining this.

I have some concern about the jump from wind SSI to gust SSI - the end result is surely very sensitive to these values. This is given some discussion in relation to Austria.

As the results in the manuscript are based upon a comparison between the seasonal forecasts and ERA5, and we use the same method to convert from windspeed to wind gust SSI for both datasets, the results will not be particularly sensitive to how we do this conversion. The only exception is in mountainous countries where we find biases in the seasonal forecast wind speeds (which are biased too low over mountain regions), which renders the bias correction less effective because the ERA5 and forecast windspeed distributions are different. This is noted in the manuscript where we emphasise that results in Austria (the country most affected by this bias) may not be reliable.

Reviewer 3

The study uses seasonal hindcasts for the preparation of an event set of European winter wind storms. The manuscript asseses storm events of ERA5 reanalysis and the GloSea hindcast using a storm severity index (SSI) which is motivated by the potential loss caused by the storm. The authors are using the UNSEEN method to analyse unprecedented wind storm events. Both, a return level analysis and the assessment of unprecedented events are done for individual countries over Europe. Finally the influence of the NAO on the storm events is investigated.

The manuscript is a revised version after first minor revision due to the necessity to better explain the novelty and added value of the study. In the current version of the manuscript, the authors highlight aspects where the manuscript goes beyond literature (line 65). I would disagree in this argumentation about novelty of the study and the contribution for further application.

Please refer to the text at the top of this document for a response regarding the novelty of the manuscript.

The manuscript does not present new ideas. As mentioned in the manuscript itself, the study of Osinski et al. (2016) already investigated an event set of European wind storms although they are not using seasonal hindacsts. It is also not the first time a seasonal hindcast is analysed focussing on wind storms (Befort et al. ,2019; Degenhardt et al., 2023). It is not the first time seasonal hincasts are used to prepare an event set of winter wind storms (Walz and Leckebusch, 2019). While the first studies are mentioned in the manuscript, it is not the case for the latter. This was already mentioned in a discussion comment (https://doi.org/10.5194/egusphere-2024-686-CC1). Since the authors are using a different seasonal prediction system and a different method to define storms and footprints as Walz and Leckebusch (2019), it is still valid to perform this study. But the advantages/disadvantages and the added value of the study has clearly to be discussed, which is completely not done so far.

We do not claim that the manuscript presents new ideas, but offers another estimation of extreme windstorm impacts, adding to the body of research already published on this topic. We already discussed several of the papers that have already done this (e.g. Osinski et al. (2016) and Lockwood et al. (2022)) and how our study differs and what is new. In a revised manuscript we will discuss Walz and Leckebusch (2019) and add an explanation of the advantages/disadvantages of the present work.

The definition of the SSI in sec. 3.2 is not properly introduced and cited in the manuscript. Klawa and Ulbrich (2003) introduce a loss index by means of the cubic exceedance of the 98th percentile of wind speed. The SSI as integrated measure of the severity of an event was introduced by Leckebusch et al. (2008). A loss index combined with population density was used by Pinto et al. (2012; https://doi.org/10.3354/cr01111). All indices use normalization by the 98th percentile which automatically reduces bias, which is not done by the SSI in the manuscript (eq. 1). It is completely valid to do it another way but the argumentation is not clear to me. The authors write it is in accordance to vendors cat models (l. 143). Is there a reference? Is this definition a better description of loss-wind relationship?

The choice of SSI definition was indeed motivated by the definition used in catastrophe models. The only reference is through personal communication with our reinsurance coauthors. It is a better definition for our purpose, in the sense that the output SSI values are directly comparable to catastrophe model output, unlike other definitions, allowing for our co-authors to evaluate the output from their catastrophe models. We will add a description of this to a revised manuscript as well as fact that the first proper SSI definition was in Leckebusch et al. (2008).

The bias adjustment of the SSI is crucial because of the sensitivity on unprecedented storms. The authors start with the same 20m/s threshold and further calculate an individual wind speed threshold for each country (with the argument of different wind distributions and wind-gust relationship). Is there any reference showing the gain of this approach in comparison to the cited literature (Klawa and Ulbrich, 2003) where the distribution (98th percentile) is directly used for normalization?

This choice was again motivated by industry practices. Reinsurance companies consider damage likely to occur when gusts are over 20 m/s. It is therefore of interest to the community to identify occasions when wind gusts will be over this threshold. Using the 98th percentile of wind speed across the fourteen different countries would not necessarily result in estimating occasions when gusts exceed 20 m/s. The 98th percentile used in Klawa and Ulbrich (2003) is an arbitrary value attained based on loss data for Germany only. We have no reason to assume the same would hold in other countries with different characteristics, for example topography, land area, length of coastline or altitude. We will explain this choice in a revised manuscript. It is also worth noting that the same bias correction method is used for ERA5 and the seasonal forecast output so the results should not be overly sensitive to the bias adjustment: if the raw wind speed from the forecast is bigger than the maximum in ERA5 it will remain so in the bias adjusted value as the same approach is used for both datasets.

In sec. 6.1 the authors compare return periods of storms (w.r.t. SSI) within ERA5 and GloSea. They shift the curves in order to match a 10y return level. Of interest is "[…] the right tail of the distribution that identifies the extreme storms, rather than the specific SSI values represented in GloSea6 (which will contain some model bias). […]" (l. 299). This shift and match to a 10y return level seems to be arbitrary. Furthermore, the SSI itself in GloSea is very important for the definition of unprecedented storms. The authors see the problem of model bias of GloSea but argue about the general agreement of the distributions of GloSea and ERA5 (l. 361). For the further usage of the results and application for end users (l. 74), choices in the methods as the way of bias adjustment or not adjustment, shift of distribution, has to be well justified.

In the calculation of unprecedented storms we do not use the shifted distribution of the SSIs, the shifted distributions are only used in Figure 6, this will be made clearer in a revised manuscript. The shift in the distributions was performed for the specific case of comparing SSI return periods from GloSea to those estimated from vendor models. In particular, how GloSea6 compared to vendor models in the shape of return period curve (especially at the extreme return periods where ERA5 is not available), and the 10-year bias correction would still preserve the slope of the GloSea6 return period curve. As we believe ERA5 gives the most accurate return period estimates we chose to shift the distributions from GloSea6 to math ERA5. We chose 10 years as it seemed like a sensible point. Below this and there are a lot of not very impactful storms, and above this the sample size of storms starts to decrease. We will clarify the methods, and motivate the choices made, in a revised manuscript.

Given the arguments about novelty of the study and further the description of the method in the context of state-of-the-art literature, I suggest to do a proper literature research and reconsider or at least explain the reasons for the way of SSI definition and bias adjustment. The chosen way and added value should be explained in the context of current literature. I suggest to reject and potentially re-submit the manuscript.

Reviewer 4

The manuscript presents an analysis of windstorm activity in seasonal forecasts - focusing on impacts, the relationship to the NAO and clustering. While the manuscript presents some interesting material, it unfortunately also includes multiple caveats that severely limit its scientific value. Therefore, I must suggest the rejection of the manuscript in its present form. Given that the other reviews already provide many details with which I agree, I only mention the most important points below.

Insufficient consideration of available peer-reviewed literature on the topic

It is clear that the authors have missed a lot of relevant literature on the topic, which may have misled them to structure the paper as it is. A revised version of the paper should include a broader view of the literature, which should help to shape the manuscript for a re-submission.

We disagree that a lot of relevant literature has been missed. Only one study directly related to the presented work that was mentioned by a reviewer was missed in the initial submission. A revised

manuscript will include this paper and we will ensure all relevant publications are properly included and discussed.

Insufficient novelty of the results

Probably related to the shortcomings described in #1, I have to concede that at the moment it is not clear to me what is new in the manuscript compared to the available literature. Several of the points that the authors claim to be novel are actually not new or only a marginal step forward from previous analysis. Please enhance.

Please see main response at the top of this document in response to this comment.

Wrong statements

Some statements such as "the NAO (…) is predictable several months in advance" are incorrect and used in a misleading way, as they suggest a deterministic predictability for the NAO on such long time scales. What the original manuscripts (Scaife et al., 2014) state, is that there is some probabilistic skill for the NAO phase in a particular MetOffice seasonal forecasting system. Please reformulate.

We did not mean to suggest deterministic predictability for the NAO (which is why we did not explicitly say deterministic). Scaife et al. 2014 showed that the GloSea5 seasonal prediction system generates forecast in which the ensemble mean has a correlation with reality of more than 0.6, and therefore contains some skill in predicting the NAO several months in advance. We will clarify this in the revised manuscript.

Overstatement of the implications of the results

In view of the above, I suggest that the authors considerable tone down the statements about the added value of the analysis performed in the manuscript.

We will carefully refine our statements on the implications and novelty of this work, based on what is truly novel and beyond the publications listed in this review, in a revised manuscript.